# Antimicrobial and antioxidant activities of neem assisted silver-modified zeolite X synthesized from kaolin

Ralph Kwakye[1]*, Grace Boakye[1], Bright Yaw Vigbedor[1], Albert Aniagyei[1], Bernard Owusu Asimeng[2], Boniface Yeboah Antwi[3] David Neglo[1], Salifu Nanga[1]

1 Basic Science Department, School of Basic and Biomedical Sciences, University of Health and Allied Sciences, Ho, Ghana, 2 Biomaterials Engineering Department, School of Engineering Sciences, University of Ghana, Legon, Ghana, 3 Institute of Industrial Research, Council for Scientific, and Industrial Research, Legon, Ghana

* kwakyer@uhas.edu.gh

## Abstract

Zeolite X was synthesized from kaolin and then modified using (*Azadirachta indica*) neem together with silver nitrate solution by ion exchange. X-ray diffraction confirmed a highly crystalline FAU framework characterised by a low angle reflection, which showed that silver incorporation did not alter the zeolite structure. Scanning Electron Microscopy (SEM) revealed a well-defined polyhedral crystal for zeolite X, whiles the Ag-zeolite X exhibited increased surface heterogeneity and agglomeration. Fourier Transform Infrared (FT-IR) and Raman Spectroscopy further confirmed framework retention, with band broadening attributed to the partial substitution of $Na^+$ by $Ag^+$ within the zeolite lattice. The antimicrobial activity of zeolite X, Ag-zeolite X, were evaluated against *Escherichia coli, Staphylococcus aureus, Klebsiella pneumoniae, Enterococcus faecalis, Candida albicans, Aspergillus niger* strains by Minimum Inhibition Concentration (MIC) and Minimum Bacterial/fungal Concentration (MBC/MFC) Assays. The antioxidant properties were evaluated using 2,2-Diphenyl-1-picrylhydrazyl (DPPH) and 2,2'-Azino-bis (3-ethylbenzothialine-6-sulfonic acid (ABTS) radical scavenging assays. The unmodified zeolite X showed weak activity (MIC, MBC, MFC > 1 mg/mL, R > 4) whilst the Ag-zeolite X showed strong antimicrobial activity (MIC = 0.5–1.0 mg/mL, MBC, MFC = 0.5–2 mg/mL, R ≤ 4) across all tested strains. Antioxidant activity assessed using DPPH and ABTS radical scavenging assays showed significantly enhanced antioxidant performance for Ag-zeolite X compared to unmodified zeolite X, particularly in the ABTS assay. These findings demonstrate that neem-assisted silver modification enhances the bioactivity of kaolin-derived zeolite X while preserving its crystalline framework, supporting its potential application in biomedical, food packaging, and water treatment systems.

**Data availability statement:** All relevant data are within the manuscript and its supporting information files.

**Funding:** The author(s) received no specific funding for this work.

**Competing interests:** The authors have declared that no competing interest exist.

## Introduction

Zeolites are crystalline microporous aluminosilicate materials. They are made up of a three-dimensional framework of $SiO_4$ and $AlO_4$ tetrahedra linked together by interconnected cavities and channels of molecular dimensions [1]. A net negative charge is generated by the substitution of $Si^{4+}$ by $Al^{3+}$ in the zeolite framework, which is compensated by substituted extra framework cations such as $Na^+$ or $Ca^{2+}$ [2]. This unique structural feature of zeolites allows them to have high ion exchange capacity, molecular selectivity, and thermal stability. It also makes them attractive for applications in adsorption, catalysis, environmental remediation, and biomedicine [3]. Among the synthetic zeolites, faujasite-type zeolite X has received considerable attention due to its large pore size, high surface area, and exceptional cation exchange capacity [4]. These properties make zeolite X particularly suitable for adsorbing bioactive metal ions such as silver. Silver exchanged zeolites (Ag-zeolites) are known to exhibit sustained antimicrobial activity through the gradual release of $Ag^+$ ions and in some cases, metallic $Ag^0$ clusters confined within the zeolite pores [5]. The antimicrobial mechanism of Ag-zeolites involves membrane disruption, binding of silver ions to thiol containing proteins, interference with enzymatic activity, and inhibition of DNA replication in microbial cells [6].

Ag-zeolites have been widely investigated as antimicrobial agents in wound dressings, coatings, food packaging, and water treatment systems [7–9]. Despite the extensive reports on the antimicrobial performance of silver loaded zeolites, most studies rely on conventional chemical reduction routes for silver adsorption, which often involve toxic reagents and lack environmental sustainability. Antimicrobial activity of Ag-zeolites is well documented but there are antioxidant properties, particularly when silver is introduced via green synthesis. Oxidative stress can delay tissue regeneration and worsen inflammation. Antioxidant activity is a critical parameter when it comes to biomedical applications, such as wound healing and implantable devices. The use of locally available natural materials in the synthesis of zeolites has gained increasing interest. This reduces production cost and improve sustainability. Kaolin synthesised zeolites offer an environmentally and economically viable alternative to commercially synthesized zeolites, particularly in developing regions. However, studies involving the combination of kaolin-based zeolite X with plant mediated silver modification remain scarce.

This study provides insights into the synergistic effects of plant mediated silver incorporation in zeolite frameworks. The findings show the potential of (*Azadirachta indica*) neem assisted Ag-zeolite X as a multifunctional material for biomedical, food packaging, and water ]treatment applications where long-lasting antimicrobial action and controlled antioxidant activity are desirable.

## Materials and methods

### Materials and reagents

Kaolin ($Al_2Si_2O_5(OH)_4$) was obtained from Wasa (5.7823°N, 2.0883°W) in the Western Region. Fresh Neem tree leaves were obtained from Dome, Ho with geographical

co-ordinates (6.62127°N, and 0.44259°E). The voucher specimens of *Azadirachta indica* were deposited at the University of Health and Allied Sciences, Institute of Traditional and Alternative Medicine (UHAS-ITAM) under accession number UHAS/ITAM/2025/LO22. The Neem plant leaves were identified by Mr. Alfred Agyemang (M.Phil. Pharmacognosy), a research fellow with the Institute of Traditional and Alternative Medicine, University of Health, and Allied Sciences. Silver nitrate (AgNO$_3$) and Sodium hydroxide (NaOH) pellets were from Sigma Aldrich (Merck) Chemical Company, Inc. UK with 99% purity, and the solvents were from Sigma Aldrich Co. Ltd, Irvine, UK, except the standard drugs. Other chemicals include DPPH (Sigma Aldrich, analytical grade, Korea), ABTS (Sigma Aldrich, analytical grade, Korea), DMSO (Sigma Aldrich, analytical grade), artesunate powder (Sigma Aldrich, analytical grade), silica gel 60 (Sigma Aldrich, analytical grade, Korea).

The test microorganisms used in this study involved *Staphylococcus aureus* (NCTC 29212), *Escherichia coli* (ATCC25922), *Klebsiella pneumoniae* (NCTC 13440), *Pseudomonas aeruginosa* (ATCC 4853)*, Enterococcus faecalis* (ATCC 19433), *Candida albicans* (ATCC 90028), and a clinical isolate *Aspergillus niger.* All the chemicals, solvents, microorganisms, distilled water, were obtained from Microbiology and Chemistry Laboratories of the School of Basic and Biomedical Sciences, University of Health and Allied Sciences, Ho. Volta Region

No permit was obtained for this study since all experiments were carried out exclusively on non living materials with the use of standard laboratory techniques and commercially sourced reagents, without the involvement of any human subjects, animals, or protected resources.

## Synthesis of zeolite X

Kaolin was pre-treated by a sieving mesh size of 45μ to remove impurities and calcined at 850°C for 3 hours to form metakaolin. 3g of activated metakaolin was mixed with 100 ml, 8 M NaOH solution and then crystallized at 100°C for 18 hours in an oven. The resulting product was filtered using filter paper and washed copiously to a pH 10, dried, and characterized.

## Silver exchange process

To prepare the plant based reducing agent, 10g of fresh neem leaves were carefully measured and washed with distilled water to remove impurities. The leaves were crushed and boiled in 100 ml of distilled water for 20 minutes at a temperature of 60–80 °C. After cooling, the extract was filtered, and the filtrate was stored in the dark for 24 hours to preserve its phytochemical integrity. The fresh neem leaf extract played a dual role in the synthesis process. First, it facilitated the green synthesis of silver nanoparticles by acting as a reducing agent, converting silver ions (Ag$^+$) into stable silver nanoparticles. Secondly, the phytochemicals in neem such as flavonoids, terpenoids, and polyphenols served as capping and stabilising agents, preventing agglomeration and ensuring the uniform dispersion of nanoparticles. This interaction not only enhanced efficiency but also helped in the adsorption of silver particles on the zeolite surface, thereby improving its antimicrobial functionality.

After the Zeolite synthesis, a 0.25 M concentration of silver nitrate (AgNO$_3$) solution was prepared. Out of the 100 mL of the crude Neem tree extract, 50 mL of the neem tree extract was mixed 50mL of AgNO$_3$ solution, stirred and heated on a hot plate magnetic stirrer set at 70 °C for 30 minutes and a color change was observed. The integration of silver (Ag$^+$) into zeolite X was performed by using the liquid ion exchange method as described by [9] with slight modifications. 3g of synthesized zeolite was weighed and introduced into the neem silver nitrate solutions. The mixture was placed on a hot plate magnetic stirrer at 30°C. The mixture was stirred for 6 hours to ensure thorough ion exchange. The obtained Ag-zeolite X was then filtered, washed with distilled water, and then dried at 100°C overnight.

## Characterization techniques

PANalytical Empyrean Powder X-ray diffractometer (PANalytical, UK) was used to assess the crystalline structure, phase purity, and framework stability of the synthesized zeolite X and the neem assisted Ag-zeolite X. Samples were finely

ground and analysed using a Cu Kα radiation diffractometer (λ = 1.5406 Å) with an operating voltage of 40 kV and current of 30 mA. Diffraction patterns were collected over a 2θ range of 5–50° with a step size of 0.02° and a scan rate of 2°min⁻¹ to resolve low angle reflections characteristic of FAU type zeolites. Zeiss EVO 500 (Zeiss, UK) scanning electron microscope was used to determine the morphology of the synthesised samples. Samples were mounted on carbon tape, sputter, coated with a thin layer of gold, and imaged at various magnifications using a field-emission SEM, The vibrational properties as well as the chemical bonds present in the samples were analyzed with a Mattson FTIR spectrometer (Mattson Instruments, UK) equipped with a ZnSe crystal plate attached to the spectrometer with a mercury cadmium telluride. FT-IR spectra were recorded in the range of 400–4000 cm⁻¹ using the KBr pellet method and Raman spectroscopy was used as a complementary technique to FT-IR to probe framework vibrations, ring breathing modes, and the metal framework interactions. Raman spectra were recorded using a suitable excitation laser wavelength to minimize fluorescence effects, with spectra collected over a range between 100 and 1200 cm⁻¹.

## MIC determination

The MICs of the synthesized zeolite samples were determined from a 2 mg/mL stock solution prepared in a 15 ml sterile falcon tube. The stock solution was prepared by weighing 10 mg of each zeolite sample and dissolving it in a 5 ml distilled water using the serial micro broth dilution method in a 96 well microtiter plates (Citotest Labware Manufacturing Co. Ltd, Jiangsu, China) per the protocol previously reported with slight modification by Clinical and Laboratory Standards Institute (CLSI) guidelines [10–12]. The minimum inhibitory concentration of the synthesised zeolites was evaluated against six selected microbial strains utilizing 96-well microtiter plates as shown in Fig 1. A concentrated solution of 0.5 mg/mL for each zeolite sample prepared by a two-fold serial dilution was conducted to achieve final concentrations ranging from 1 mg/mL to 0.01 mg/mL. Each well was filled with 100 µL in a double-strength Mueller Hinton broth (Oxoid Limited, United Kingdom), followed by the addition of 100 µL of the corresponding zeolite solution. Wells 11 and 12 were designated as positive (broth + organism) and negative (broth only) controls, respectively. Tetracycline (1 mg/mL) and Nystatin (1 mg/mL) were used as positive control antibiotics for bacterial and fungal strains. The test organisms were cultured overnight at 37 °C in Mueller Hinton broth, and their optical density was adjusted to 1 × 10⁶ CFU/mL, standardized to 0.5 McFarland. Subsequently, 100 µL of this suspension was added to each well, and the plates were incubated at 37 °C for 48 hours for both bacterial and fungal strains. The MIC values were determined using tetrazolium chloride (TTC, 0.1% w/v dye),

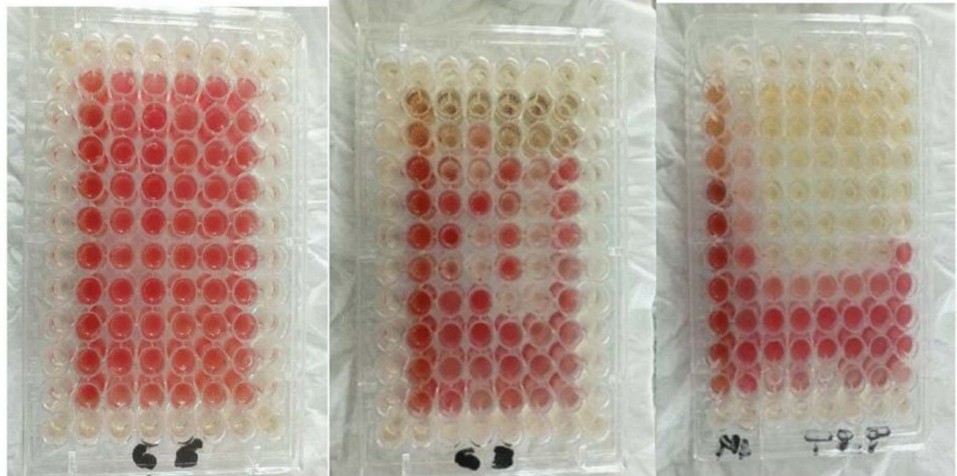

**Fig 1. Antimicrobial activity of A: zeolite X B: Ag-zeolite X C: controls for all organisms.**

with the MIC defined as the lowest concentration that inhibited the color change from colorless/light yellow. All tests were performed in duplicate to ensure accuracy.

## MBC and MFC determinations

The investigation into the bactericidal and fungicidal effects of Ag-zeolite X involved the assessment of Minimum Bactericidal Concentration (MBC) and Minimum Fungicidal Concentration (MFC) as per the methodology of Öztürk & Ercisli. Aliquots from each well of the MIC assay were transferred to sterilized nutrient agar for bacterial analysis and Sabouraud Dextrose Agar (SDA) for fungal analysis. The inoculated plates were incubated at 37°C for 48 hours, after which microbial growth was evaluated. The MBC and MFC were recorded at the lowest concentration at which no visible growth was observed, indicating a 99.9% reduction in microbial cell numbers [13].

## Determination of antioxidant activity

The determination of total antioxidant activity was conducted through the application of spectrophotometric methods, specifically DPPH and ABTS assays.

## DPPH radical scavenging activity assay

The assessment of DPPH scavenging activities for Ag-zeolite X was conducted by the protocol modified by [14,15] specifically adjusting the incubation duration. The DPPH radical scavenging assay (DPPH RSA) was utilized to evaluate the antioxidant efficacy of the zeolite samples by measuring their ability to neutralize free radicals. A 0.1 mM DPPH solution was prepared by mixing 1 mL of a 1 M DPPH stock solution with 9 mL of methanol. In microplate wells, 40 μL of a 0.01, 0.05, 0.25, 0.5 and 1 mg/mL zeolite and Ag-zeolite X samples were added to 160 μL of the DPPH solution. The mixture was then incubated in the dark at room temperature for one hour, and absorbance was measured at 517 nm using a UV spectrophotometer (Drawell DNM-9602 microplate reader). The radical scavenging capacity was compared to ascorbic acid (0.01, 0.05 0.25, 0.5 and 1 mg/mL) as a positive control. The entire experimental procedure was replicated three times, and the radical scavenging activity was expressed as a percentage of the control based on the reduction in absorbance.

$$\% \text{ inhibition} = \frac{(AC - At)}{AC} \times 100\%$$

Where Ac was the absorbance of the control (in which the same volume of Methanol was used in place of the zeolites) and at was the absorbance in the presence of the zeolite.

## ABTS radical scavenging activity assay

The evaluation of the ABTS free radical scavenging ability Ag-zeolite X was conducted following the methodology established by [16] and further referenced by [14] with slight modifications. A 0.1 mM ABTS solution was created by mixing 1 ml of a 1 M ABTS stock solution with 9 ml of methanol. Subsequently, 40 μl of a zeolite and Ag-zeolite samples at a concentration of 0.01 0.05, 0.1, 0.25, 0.5, 1 mg/ml was combined with 160 μl of the ABTS solution in 96-well microplates. The mixture was then incubated in the dark at room temperature for one hour. This experimental procedure was replicated three times. Absorbance readings were taken at 734 nm using a UV spectrophotometer (Drawell DNM-9602 microplate reader). Ascorbic acid 0.01,0.05, 0.25, 0.5 and 1 mg/mL) served as the positive controls, while the ABTS assay functioned as the negative controls. The radical scavenging activity was determined as a percentage of the control, calculated based on the reduction in absorbance.

$$\% \text{ inhibition} = \frac{(Ac - At)}{Ac} \times 100$$

The absorbance of the negative control sample was denoted as Ac, while the absorbance of the zeolite and Ag-zeolite ABTS radical solution represented as At

## Results and discussion

### XRD of zeolite X and silver exchanged zeolite X

The XRD pattern exhibits distinct peaks at 6.1°, 11.7°, 15.5°, 23.3°, 26.5°, and 31.2° (2θ), correspond to the cubic FAU framework (JCPDS Card No. 38–0237) reported by [17]. These peaks correspond to cubic faujasite structure confirming the successful synthesis of zeolite type X. The presence of multiple sharp diffraction peaks indicates high crystallinity, which is a crucial property for zeolites used in catalysis, adsorption, and ion exchange applications. The peak at 6.1° is particularly significant, as it is a major indicator of large pore openings typical of faujasite structures which facilitates high silver loading and enhanced antimicrobial potential as reported by [18]. The high intensity of these peaks, especially at low angles, confirms the presence of a large unit cell. The positions of the peaks were the same after the silver ion exchange, confirming the structural integrity of the zeolite framework. There were no additional peaks or shifts in the peaks of the neem assisted Ag-zeolite X as compared to the zeolite X. This is an indication of the adsorption of the silver ions or silver nanoparticles into the zeolite framework structure without forming any additional crystalline phase as shown in Fig 2. The preservation of multiple high-intensity reflections after modification indicates minimal framework collapse or amorphization during silver incorporation, aligning with findings by many authors. This shows that the silver exchange process did not compromise the structural integrity of the unmodified zeolite X.

### SEM of zeolite and silver exchanged zeolite X

The SEM images showed a more distinct difference in morphology between the unmodified zeolite X (A) and the Ag-zeolite X (B). This may be due to the silver ion exchange. The particles in the zeolite X are uniform in shape and size averaging around 1–3 μm as reported by [19]. They also exhibit sharp edged polyhedral crystals which are characteristics of synthetic faujasite type Zeolites. In comparison, the particles in the silver loaded zeolite X appear more agglomerate, less crystalline and with a less distinct shape. The agglomeration may be due to the adsorption of silver [20]. High

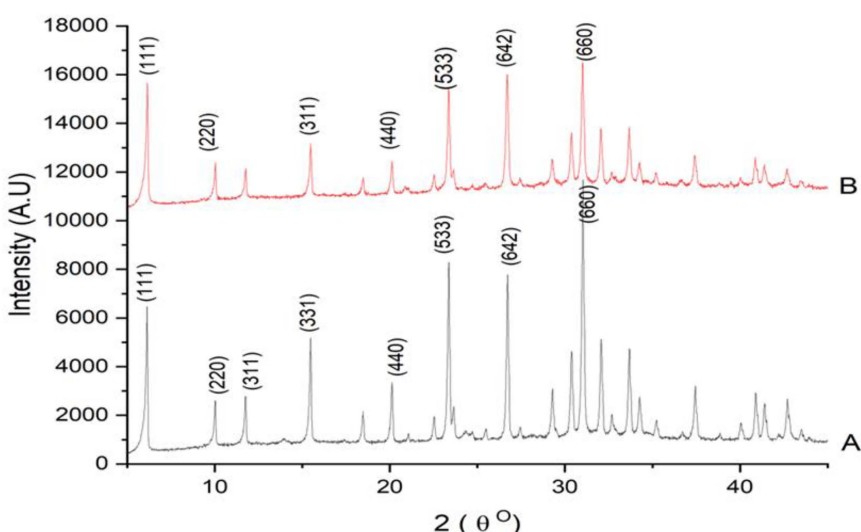

**Fig 2. XRD pattern of A: zeolite X B: Ag-zeolite X showing characteristic faujasite peaks and preservation of structural integrity after silver incorporation.**

crystallinity was observed in zeolite X, which is very good for ion exchange [21], but there was partial amorphization in the neem assisted silver loaded zeolite X [22]. The difference in the morphology of the Ag-zeolite X indicates a successful silver ion exchange [23]. These changes can enhance antimicrobial activity due to the increased roughness and potential formation of silver nanoparticles [24] as shown in Fig 3(A) and 3(B).

## FT-IR of zeolite X and silver exchanged zeolite X

From the Fourier Transform Spectroscopy, a broad absorption peak around 400–1600 cm$^{-1}$ was observed for both samples in this region. This was mostly due to the asymmetric stretching vibrations of the Si–O–Si and Si–O–Al bonds that are found in the tetrahedral framework of zeolites [25]. A weaker peak at 790 cm$^{-1}$ corresponding to the symmetric stretching of internal tetrahedral in zeolites was also observed. Peaks ranging from 460 cm to 550 cm$^{-1}$ that indicates bending vibrations due to Si-O or Al-O characteristic of zeolite frameworks were also observed [26]. In comparison there was broadening in the key adsorption peaks. This indicates lattice distortions because of the adsorption of the Ag$^+$. These ions were exchanged with the Na$^+$ ions in the Zeolite framework [27] as shown in Fig 4. There was also a high absorbance of Ag$^+$ around 800–1000 cm$^{-1}$ meaning that there was high interaction between the Ag$^+$ ions and oxygen atoms in the framework structure of the zeolite. The broad peak observed in the neem assisted Ag-zeolite X indicates structural disorders due to the absorption of silver ions [28]. These observed modifications in the FT-IR spectra are attributed to the electronic environment of the framework tetrahedra because of the ion exchange. The electrostatic interaction between Ag$^+$ and negatively charged oxygen atoms in the aluminosilicate network and the Ag–O coordination led to a new or shifted vibrational modes [29]. All these are consistent with previously reported modifications in ion-exchanged zeolites. This supports the hypothesis that silver ions if effectively integrated into a zeolite framework may enhance the material's antimicrobial properties [30].

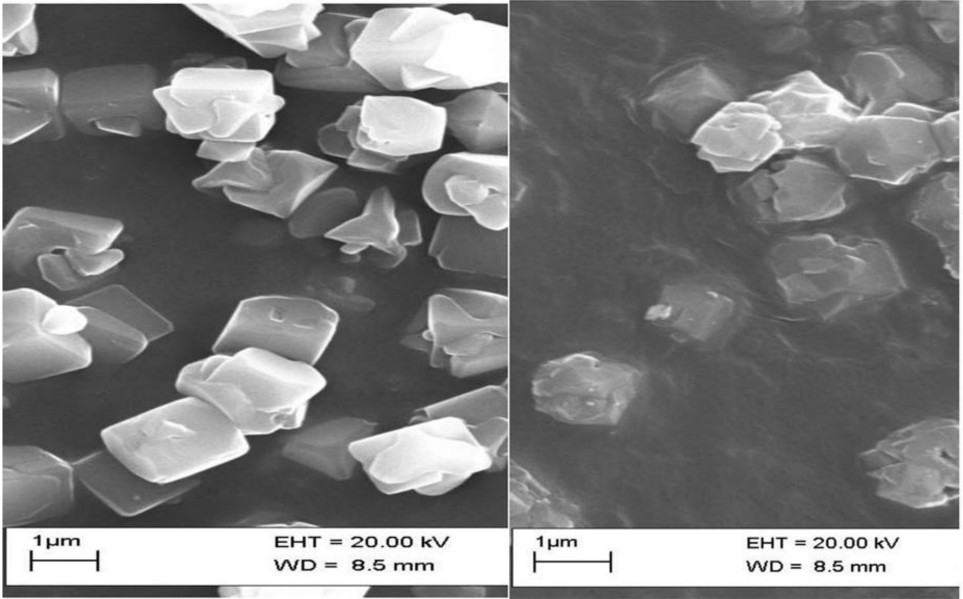

**Fig 3. SEM image of A: zeolite X displaying sharp polyhedral crystals B: Ag- zeolite X showing agglomerated less crystalline structure after ion-exchange.**

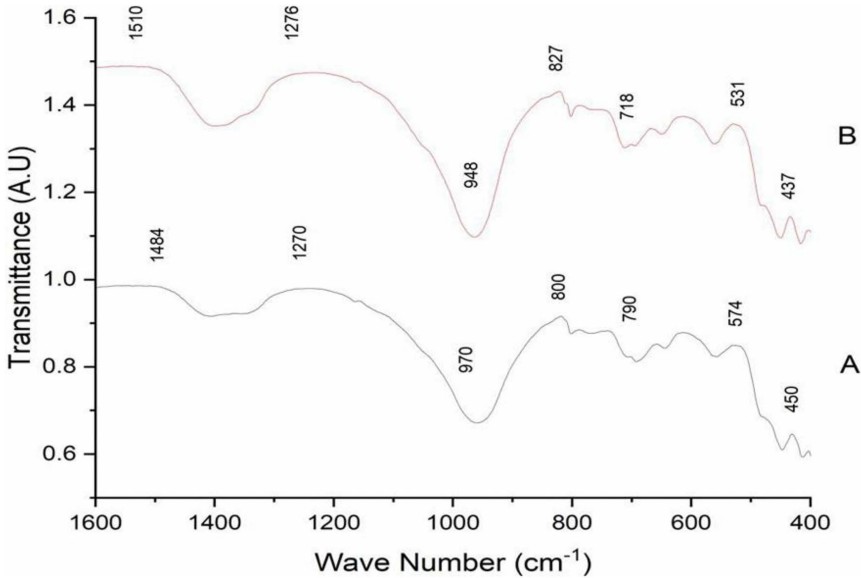

**Fig 4. FT-IR of A: zeolite X B: Ag-zeolite X showing broadening of peaks after silver adsorption.**

## Raman spectra of zeolite and silver exchanged zeolite X

The Raman spectra of zeolite X (A) and neem assisted Ag-zeolite X (B) reveal distinct vibrational features characteristic of the aluminosilicate framework and provide clear evidence of structural perturbation following the Ag+ incorporation Fig 5. In both spectra, the dominant bands appear in the 540–600 cm⁻¹ region, which is well established as the region for framework T–O–T (T = Si or Al) bending and ring-breathing modes in faujasite-type zeolites [20,26,31–35].

In the zeolite X (A), the most intense Raman band is observed at 545 cm⁻¹, accompanied by secondary features near 563 cm⁻¹ and 586 cm⁻¹. The strong band at 545 cm⁻¹ is characteristic of symmetric breathing vibrations of double six-membered rings (D6R), a structural unit fundamental to the FAU topology [20]. The bands at higher wavenumbers 560–590 cm⁻¹ are commonly attributed to asymmetric T–O–T bending modes and reflect the distribution of Si–O–Al linkages within the aluminosilicate framework [31,32]. The relative sharpness and high intensity of these peaks indicate a well-ordered crystalline framework with minimal structural distortion.

Upon silver loading Ag-zeolite X, notable spectral modifications were observed. The principal Raman band shifts from 545 cm⁻¹ in the zeolite X to 544 cm⁻¹, while the band near 563 cm⁻¹ in zeolite X shifts to approximately 562 cm⁻¹ in the Ag-zeolite X and shows a marked decrease in intensity. Similarly, the feature at 586 cm⁻¹ in the zeolite X shifts to 585 cm⁻¹ and becomes broader and less intense. These shifts and intensity reductions are indicative of framework perturbations induced by Ag+ incorporation into the zeolite cavities [32].

Such spectral changes can be attributed to several concurrent effects. First, ion exchange of Na+ by Ag+ modifies the local electrostatic environment of the framework oxygen atoms, weakening T–O bonds and lowering their vibrational frequencies [23]. Second, partial coordination of Ag+ with framework oxygen atoms or extra-framework positions can distort the D6R units, leading to increased structural disorder and peak broadening [34]. The reduced overall Raman intensity observed for the Ag-zeolite X further suggests partial reduction of framework polarizability due to the presence of heavy silver species, which can suppress vibrational modes through mass and electronic effects.

Notably, no new intense bands attributable to crystalline silver oxide or metallic silver clusters were observed in the measured spectral window, suggesting that silver is present as dispersed ionic species or small clusters rather than large

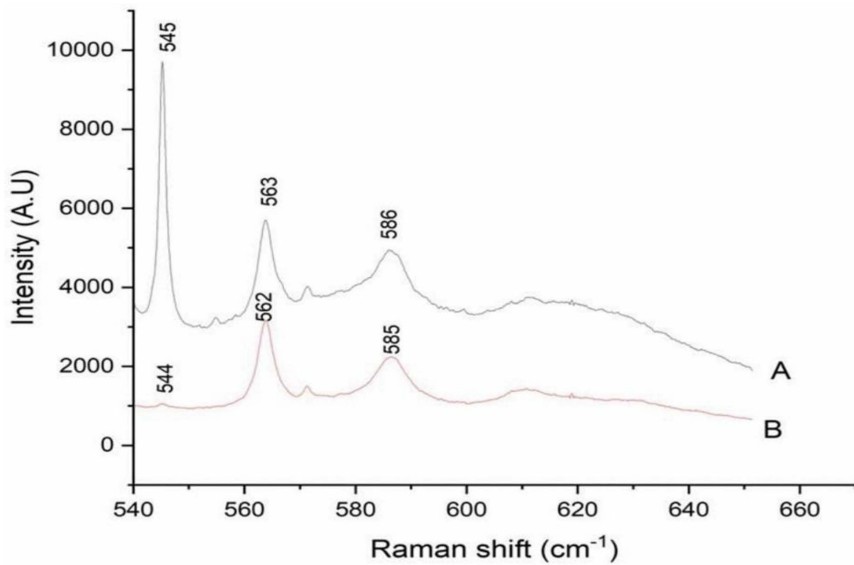

**Fig 5. Raman spectra of A: zeolite X B: Ag-zeolite X showing broadening of peaks after silver adsorption.**

crystalline aggregates. This observation is consistent with previous Raman and XRD studies of Ag-exchanged zeolites, where well-dispersed $Ag^+$ species do not generate distinct Raman active modes in the mid frequency region [5,31–36].

## Antimicrobial activity

The antimicrobial activity of the synthesized zeolites was evaluated by investigating their bioactivity before and after silver incorporation into the zeolite X framework on microbial inhibition. The microbial activity of zeolite X, neem assisted Ag-zeolite X together with Tetracycline and Nystatin as controls were tested against Gram-positive bacteria (*Staphylococcus aureus* and *Enterococcus faecalis*), Gram-negative bacteria (*Escherichia coli* and *Klebsiella pneumoniae*), as well as fungal and yeast strains (*Aspergillus Niger* and *Candida albicans*). Tetracycline and nystatin were used as standard antibacterial and antifungal controls, respectively. Antimicrobial efficacy was determined using minimum inhibitory concentration (MIC) assays, complemented by minimum bactericidal and fungicidal concentration (MBC/MFC) determinations, with the activity ratio (R = MBC/MIC) was used to differentiate between bacteriostatic/fungistatic and bactericidal/fungicidal effects. The unmodified zeolite X consistently exhibited weak antimicrobial activity against all tested microorganisms. As shown in Table 1. MIC values for zeolite X were greater than 1 mg/mL for both bacterial and fungal strains, with corresponding MBC and MFC values also exceeding 1 mg/mall The calculated activity ratios (R > 4) indicated that zeolite X exerted only bacteriostatic or fungistatic effects rather than bactericidal or fungicidal action. This weak antimicrobial performance is expected, as zeolite X is an aluminosilicate material that do not have any intrinsic antimicrobial agents and only functions as an inert adsorption matrix. Similar findings have been reported in earlier studies, where unmodified zeolites showed minimal antimicrobial activity unless modified with bioactive metal ions such as silver, copper, or zinc [6,9,35,36]. In contrast, neem assisted Ag-Zeolite X demonstrated strong and broad-spectrum antimicrobial activity against all tested microorganisms. The MIC and MBC/MFC values for Ag-Zeolite X, presented in Table 1, ranged from 0.5 to 1.0 mg/mL and 0.5 to 2.0 mg/mL, respectively, depending on the microbial strain. The corresponding activity ratios (R ≤ 4) confirm that Ag-zeolite X exerted bactericidal and fungicidal effects rather than just inhibiting microbial growth. These results clearly demonstrate that silver incorporation together with neem significantly enhanced the antimicrobial efficacy of zeolite X. Among the bacterial strains tested, *E. coli*, *E. faecalis*, and *K. pneumoniae* exhibited MIC values of 0.5 mg/mL when

**Table 1. Minimum Inhibitory Concentration (MIC), Minimum Bactericidal/ Fungal concentration (MBC/ MFC), and activity ratio, R = MBC/MIC) of zeolite X, Ag-Zeolite X, tetracycline, and nystatin against tested microbial strains.**

| Organisms | Zeolite X | | | Ag-zeolite X | | | Tetracycline | | |
|---|---|---|---|---|---|---|---|---|---|
| | MIC | MBC | R | MIC | MBC | R | MIC | MBC | R |
| **Bacteria** | | | | | | | | | |
| *E. coli* | >1 | >1 | >4[bs] | 0.5 | 0.5 | 1[bc] | 0.313 | 0.63 | 2.[bc] |
| *E. faecalis* | >1 | >1 | >4[bs] | 0.5 | 2 | 4[bc] | 0.0156 | 0.0156 | 1[bc] |
| *K. pneumonia* | >1 | >1 | >4[bs] | 0.5 | 1 | 2[bc] | 0.0156 | 0.063 | 4[bc] |
| *S. aures* | >1 | >1 | >4[bs] | 1 | 2 | 2[bc] | 0.0156 | 0.063 | 4[bc] |
| **Fungus** | | | | | | | **Nystatin** | | |
| *A. Niger* | >1 | >1 | >4[fs] | 0.5 | 1 | 2[bc] | 0.5 | 1 | 2[fc] |
| **Yeast** | | | | | | | | | |
| *C. albican* | >1 | >1 | >4[fs] | 1 | 2 | 2[bc] | 0.125 | 0.25 | 2[fc] |

[bc]: bactericidal, [bs]: bacteriostatic, [fc]: fungicidal, [fs]: fungistatic, [R ≤ 4 = bactericidal or fungicidal R > 4 = bacteriostatic or fungistatic].

treated with Ag-zeolite X, with MBC values ranging from 0.5 to 2.0 mg/mL (Table 1). These results indicate high susceptibility of both Gram-negative and Gram-positive bacteria to the silver modified zeolite. *Staphylococcus aureus* displayed slightly reduced sensitivity, with an MIC of 1.0 mg/mL and an MBC of 2.0 mg/mL, as shown in Table 1. This reduced susceptibility may be attributed to the thick peptidoglycan layer characteristic of Gram-positive bacteria, which can limit diffusion and delay the penetration of antimicrobial agents [6,35]. That not withstanding, the calculated R value of 2 confirms that Ag-zeolite X retains bactericidal activity against *S. aureus*.

The antifungal activity of Ag-Zeolite X followed a similar trend. As shown in Table 1, *Aspergillus Niger* exhibited an MIC of 0.5 mg/mL and an MFC of 1.0 mg/mL, while *Candida albicans* showed an MIC of 1.0 mg/mL and an MFC of 2.0 mg/mall These findings demonstrate that Ag-zeolite X is effective against both filamentous fungi and yeast. The fungicidal activity of silver-modified zeolites has been previously attributed to silver-induced disruption of fungal cell membranes, interference with mitochondrial respiration, and inhibition of key enzymatic pathways [36,37].

A comparative assessment with standard antimicrobial agents revealed that, although tetracycline and nystatin exhibited lower MIC values than Ag-zeolite X (Table 1), the antimicrobial performance of Ag-zeolite X was statistically comparable to tetracycline. Differences in minimum inhibitory concentration (MIC) among treatments were statistically significant ($H(2) = 10.23$, $p = 0.006$). Post hoc Dunn's tests showed that the MIC of unmodified zeolite X was significantly higher than that of Ag–zeolite X ($p = 0.004$) and tetracycline ($p = 0.001$). No significant difference was observed between Ag–zeolite X and tetracycline ($p = 0.317$). Similarly, minimum bactericidal/fungicidal concentration (MBC/MFC) values differ significantly among treatments ($H(2) = 9.45$, $p = 0.009$). Pairwise comparisons indicated that unmodified zeolite X differed significantly from Ag–zeolite X ($p = 0.006$) and tetracycline ($p = 0.002$), whereas no significant difference was detected between Ag–zeolite X and tetracycline ($p = 0.284$). Based on MBC/MIC ratios (R ≤ 4), Ag–zeolite X exhibited bactericidal and fungicidal activity against all tested strains, whereas unmodified zeolite X was bacteriostatic and fungistatic (R > 4). These results further confirm the strong antimicrobial efficacy of Ag-zeolite X relative to conventional antibiotics.

The enhanced antimicrobial performance of Ag-zeolite X can be attributed primarily to the surface morphology and particle aggregation of silver ions and silver nanoparticles immobilized within the zeolite framework as shown in the SEM image. Zeolite X functions as a controlled release carrier, enabling sustained liberation of Ag⁺ ions into the surrounding environment. Silver ions exert a multi-target antimicrobial mechanism, including disruption of cell membrane integrity, binding to thiol (–SH) groups in proteins, inhibition of essential metabolic enzymes, and interference with DNA replication [4,8,38]. This multifaceted mode of action explains the bactericidal and fungicidal behaviour observed in Table 1 and reduces the likelihood of resistance development. In addition, the neem-assisted synthesis route contributed to the

enhanced antimicrobial activity of Ag-zeolite X. Neem derived phytochemicals, including flavonoids, terpenoids, and polyphenols, can function as stabilizing agents for silver nanoparticles and may also possess intrinsic antimicrobial properties. The synergistic interaction between silver species and neem phytochemicals may enhance microbial membrane permeability and facilitate silver ion uptake, further strengthening antimicrobial efficacy [38].

The observation that Ag-Zeolite X exhibited bactericidal activity against both Gram-negative and Gram-positive bacteria (as evidenced by R ≤ 4 values in Table 1) suggests that the outer membrane of Gram-negative bacteria does not significantly impede the biocidal action of silver ions. The slightly higher MIC values observed for *S. aureus* and *C. albicans* may be attributed to the structural complexity of their cell walls, which can partially restrict silver ion diffusion [6,35–37]. Despite this, Ag-zeolite X maintained strong antimicrobial activity across all tested strains. The data presented in Table 1 demonstrate that silver exchange transforms zeolite X from a biologically inert material into a highly effective broad spectrum antimicrobial agent. The substantial reduction in MIC and MBC/MFC values upon silver incorporation confirms that silver is the primary antimicrobial component, while the zeolite framework plays a critical role in stabilizing and regulating silver release. These findings are consistent with previous studies on silver exchanged Faujasite zeolites [6,9,36–38] and strongly support the potential application of neem-assisted Ag-zeolite X in biomedical devices, wound dressings, food packaging, and water purification systems.

Differences in minimum inhibitory concentration (MIC) among treatments were statistically significant ($H(2) = 10.23$, $p = 0.006$). Post hoc Dunn's tests showed that the MIC of unmodified zeolite X was significantly higher than that of Ag–zeolite X ($p = 0.004$) and tetracycline ($p = 0.001$). No significant difference was observed between Ag–zeolite X and tetracycline ($p = 0.317$). Similarly, minimum bactericidal/fungicidal concentration (MBC/MFC) values differ significantly among treatments ($H(2) = 9.45$, $p = 0.009$). Pairwise comparisons indicated that unmodified zeolite X differed significantly from Ag–zeolite X ($p = 0.006$) and tetracycline ($p = 0.002$), whereas no significant difference was detected between Ag–zeolite X and tetracycline ($p = 0.284$). Based on MBC/MIC ratios (R ≤ 4), Ag–zeolite X exhibited bactericidal and fungicidal activity against all tested strains, whereas unmodified zeolite X was bacteriostatic and fungistatic (R > 4).

## Antioxidant properties

The DPPH radical scavenging assay (DPPH RSA) and ABTS radical scavenging assay (ABTS RSA) were used to assess the antioxidant activities of zeolite X, Ag- zeolite X and ascorbic acid (Vitamin C) as control Fig 6A and B. The concentrations range is from 0.05–0.25 mg/ml.

The radical scavenging activity of all the samples increases with an increase in concentration in the DPPH assay. The silver modified Zeolite X exhibited higher inhibition in the DPPH assay compared to the unmodified zeolite, except at the concentration 0.05 mg/ml, but the control, vitamin C was superior to all Fig 6B.

A Two-way Anova test showed a significant difference between the sample type and concentration ($p < 0.0001$) as against the inhibition ($p < 0.0001$). This implies that antioxidant activity depends on material type together with its concentration.

Turkey HSD analysis further showed that Vitamin C was significantly more active ($p < 0.05$) than the silver modified zeolite X and the unmodified zeolite X. The Ag-zeolite X consistently showed higher inhibition, compared to the zeolite X in the DPPH assay. The improved scavenging activity showed by the silver modified zeolite X in the DPPH assay compared to the unmodified zeolite may be attributed to an enhance surface reactivity and electron transfer by the silver ions [2].

Similarly, in the ABTS radical scavenging assay Fig 5B, the percentage inhibition of the vitamin C, Ag-zeolite X and the unmodified zeolite X were determined across the tested concentrations 0.5–0.25 mg/ml. The percentage inhibition increases as the concentration of the tested samples increases. Vitamin C showed the highest radical scavenging activity followed by Ag-zeolite X and zeolite X.

A two-way Anova test also revealed that both the sample type ($F(2, 30) = 359.61$, $p < 0.0001$) and concentration ($F(4, 30) = 18.02$, $p < 0.0001$) had significant effect on ABTS radical scavenging. There was also significant effect of interaction between various samples and their concentration ($F(8, 30) = 8.30$, $p < 0.0001$). A Turkey HSD post-hoc comparison really confirmed the

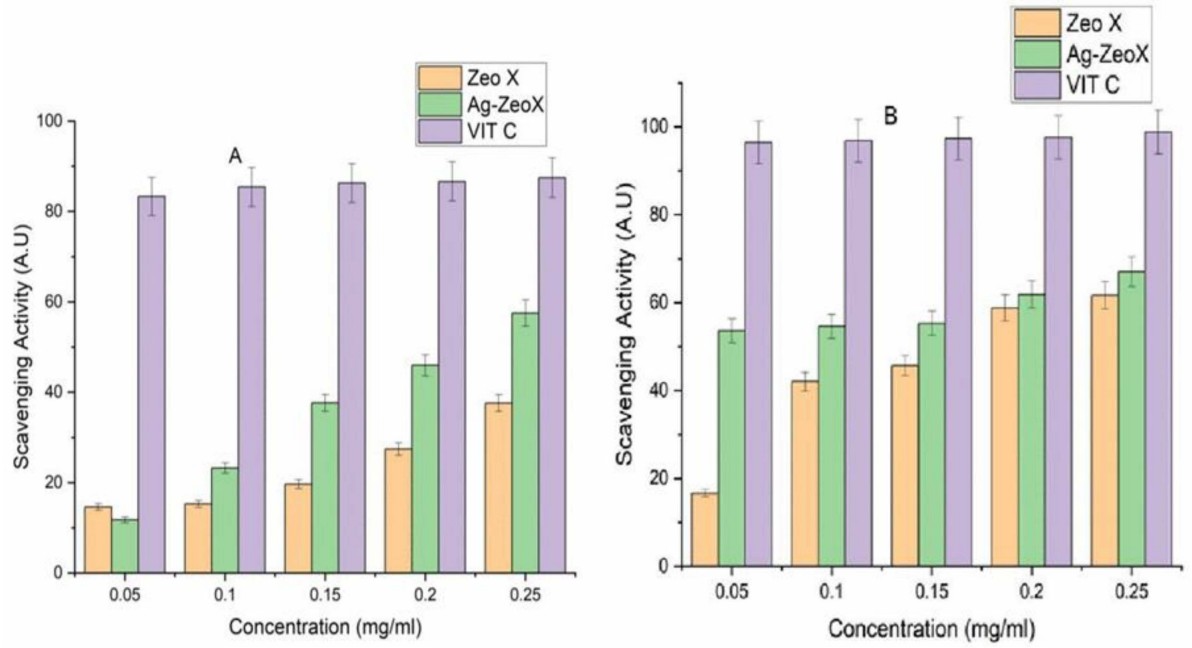

**Fig 6. A: DPPH B: ABTS scavenging activity of synthesized zeolite X, Ag-zeolite X and Vit C.**

superiority of the control, vitamin C ($p<0.05$) inhibition as against the silver modified zeolite and the unmodified zeolite across all concentrations. The scavenging activity of the silver modified zeolite X was more enhanced C ($p<0.05$) as in the ABTS radical scavenging assay as compared to the unmodified zeolite X. These findings suggest that neem assisted silver modified zeolite X effectively enhances electron transfer and hydrogen atom donation mechanisms which are central to ABTS radical scavenging [3].

This study agrees with the findings that metal exchanged zeolites show high redox and catalytic properties due to their enhance charge transfer kinetics [3–4]. The present findings suggest the silver loaded zeolite is a good material for biomedical and environmental applications requiring controlled antioxidant activity such as wound healing, antimicrobial coatings, and drug delivery matrices where oxidation stress modulation is required.

The DPPH and the ABTS analysis showed that silver loading really improved the antioxidant efficiency of the zeolite X even though the percentage inhibition was higher in the ABTS assay. This may be due to the fact the silver ions might interfere with the hydrogen atom or electron transfer processes in the DPPH assay, leading to a decrease in the scavenging activity. [39]. DPPH radical scavenging assay relies primarily on the hydrogen atom transfer (HAT) mechanism and is quite selective [40–43]. Silver ions, while efficient in electron transfer, do not readily participate in hydrogen atom donation, a critical pathway in the DPPH assay [41].

The silver modification really enhanced the electron transfer and the radical stabilization mechanisms confirming its synergetic role for metal exchange in optimizing zeolite-based antioxidant systems.

## Conclusion

Neem assisted silver modified zeolite X (Ag-zeolite X) was successfully synthesized. The modified silver zeolite X exhibited a broad-spectrum antimicrobial and antioxidant activity. The synthesized neem assisted Ag-zeolite X showed significant bactericidal, fungicidal effects as well as enhanced ABTS radical scavenging. The dual functionality of this synthesized neem assisted silver modified zeolite supports its potential applications in biomedical devices such as wound dressings, bandages, surgical masks and in food packaging and water purification.

## Supporting information

**S1 File. Fig 5. Data 2.**
(DOCX)

## Acknowledgments

We thank Dr. Bernard Amoako-Hene of the liberal department for editing the manuscript and all the technicians and staff of the School of Basic and Biomedical Sciences for their dedication. We also thank the Government of Ghana for the Book and Research allowance.

## Author contributions

**Conceptualization:** Ralph Kwakye.

**Data curation:** Boniface Yeboah Antwi, Salifu Nanga.

**Formal analysis:** Bright Yaw Vigbedor, Albert Aniagyei, Bernard Owusu Asimeng, Salifu Nanga.

**Investigation:** David Neglo.

**Methodology:** David Neglo.

**Supervision:** Bernard Owusu Asimeng.

**Validation:** Albert Aniagyei, Boniface Yeboah Antwi.

**Writing – original draft:** Ralph Kwakye, Grace Boakye.

**Writing – review & editing:** Bright Yaw Vigbedor.

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
