## [Decision Letter · Decision Letter 0]

9 Dec 2025

Dear Dr. KWAKYE,

Thank you for submitting your manuscript to PLOS ONE. After careful consideration, we feel that it has merit but does not fully meet PLOS ONE’s publication criteria as it currently stands. Therefore, we invite you to submit a revised version of the manuscript that addresses the points raised during the review process.

We look forward to receiving your revised manuscript.

Kind regards,

Safdar Ali Amur

Academic Editor

PLOS ONE

**Journal Requirements:**

1. When submitting your revision, we need you to address these additional requirements. Please ensure that your manuscript meets PLOS ONE's style requirements, including those for file naming. The PLOS ONE style templates can be found at https://journals.plos.org/plosone/s/file?id=wjVg/PLOSOne_formatting_sample_main_body.pdf and https://journals.plos.org/plosone/s/file?id=ba62/PLOSOne_formatting_sample_title_authors_affiliations.pdf 2. In your Methods section, please include information about voucher specimens including the name and qualification of the expert providing verification of the sample identification (Azadirachta indica). 3. In your Methods section, please provide additional information regarding the permits you obtained for the work. Please ensure you have included the full name of the authority that approved the field site access and, if no permits were required, a brief statement explaining why. 4. We suggest you thoroughly copyedit your manuscript for language usage, spelling, and grammar. If you do not know anyone who can help you do this, you may wish to consider employing a professional scientific editing service.  The American Journal Experts (AJE) (https://www.aje.com/) is one such service that has extensive experience helping authors meet PLOS guidelines and can provide language editing, translation, manuscript formatting, and figure formatting to ensure your manuscript meets our submission guidelines. Please note that having the manuscript copyedited by AJE or any other editing services does not guarantee selection for peer review or acceptance for publication.  Upon resubmission, please provide the following: The name of the colleague or the details of the professional service that edited your manuscript A copy of your manuscript showing your changes by either highlighting them or using track changes (uploaded as a *supporting information* file) A clean copy of the edited manuscript (uploaded as the new *manuscript* file) 5. We note that your Data Availability Statement is currently as follows: All relevant data are within the manuscript and its supporting information files. Please confirm at this time whether or not your submission contains all raw data required to replicate the results of your study. Authors must share the “minimal data set” for their submission. PLOS defines the minimal data set to consist of the data required to replicate all study findings reported in the article, as well as related metadata and methods (https://journals.plos.org/plosone/s/data-availability#loc-minimal-data-set-definition). For example, authors should submit the following data: - The values behind the means, standard deviations and other measures reported;- The values used to build graphs;- The points extracted from images for analysis. Authors do not need to submit their entire data set if only a portion of the data was used in the reported study. If your submission does not contain these data, please either upload them as Supporting Information files or deposit them to a stable, public repository and provide us with the relevant URLs, DOIs, or accession numbers. For a list of recommended repositories, please see https://journals.plos.org/plosone/s/recommended-repositories. If there are ethical or legal restrictions on sharing a de-identified data set, please explain them in detail (e.g., data contain potentially sensitive information, data are owned by a third-party organization, etc.) and who has imposed them (e.g., an ethics committee). Please also provide contact information for a data access committee, ethics committee, or other institutional body to which data requests may be sent. If data are owned by a third party, please indicate how others may request data access. 6. If the reviewer comments include a recommendation to cite specific previously published works, please review and evaluate these publications to determine whether they are relevant and should be cited. There is no requirement to cite these works unless the editor has indicated otherwise. 

Reviewers' comments:

**Comments to the Author**

1. Is the manuscript technically sound, and do the data support the conclusions?

Reviewer #1: Yes

Reviewer #2: Yes

2. Has the statistical analysis been performed appropriately and rigorously?

Reviewer #1: Yes

Reviewer #2: Yes

3. Have the authors made all data underlying the findings in their manuscript fully available?

Reviewer #1: Yes

Reviewer #2: Yes

4. Is the manuscript presented in an intelligible fashion and written in standard English?

Reviewer #1: Yes

Reviewer #2: Yes

**Reviewer #1:**  1. introduction lacks a novelty statement. The last paragraph of the introduction is not well fit here. It must be revised to focus on the aim and novelty of the work.

2. Throughout MS, make the writing style of silver ions or silver uniform.

3. Follow the correct scientific writing of tested microbes.

4. Remove repetition of DPPH and NBTS chemicals in materials and methods.

5. For metakaolin: give accurate oven temp. and how filtration was done.

6. neem leaves (Azadirachta indica): the scientific name should be used firstly in the introduction part, then either write the scientific name or the English name.

7. Explain in detail the methodology of characterization techniques.

8. In MIC, how were the stock solutions were made?

9. For XRD results, add more supportive literature and provide JCPDS card numbers of materials.

10. Add more peak numbers in the FTIR and Raman results discussion and justify with literature.

11. Correct the Table 1 format. Antibacterial results should be discussed in detail and also described in correlation with SEM data.

12. Correct the format style of the p-value throughout the MS.

**Reviewer #2:**  1. Revise the abstract to include key results of synthesis.

2. introduction should be revised to justify the study title and include a novelty statement.

3. Make uniformity to write chemical names, plant names, and tested organisms.

4. Provide a detailed method of characterization.

5. Mostly, temp/time in many experiments is not given.

6. Results need to be strongly discussed in detail and supportive literature.

7. Add peak numbers in XRD and FTIR graphs. Add scale bar in SEM images.

8. Supply the antimicrobial visual images in the main article.

9. Overall, the study is OK, but novelty is not discussed anywhere, and results are not encouraged to stand alone as worthy of work.

**Do you want your identity to be public for this peer review?** For information about this choice, including consent withdrawal, please see our Privacy Policy

Reviewer #1: No

Reviewer #2: No

---

## [Author Response · Author response to Decision Letter 1]

9 Jan 2026

Response to Reviewers

Manuscript Title:

Antimicrobial and Antioxidant Activities of Neem-Assisted Silver-Modified Zeolite X Synthesized from Kaolin

Manuscript ID: [PONE-D-25-58884]

We thank the Editor and Reviewers for their constructive comments, which have greatly improved the clarity, scientific rigor, and presentation of our manuscript. All comments have been carefully addressed, and the manuscript has been extensively revised accordingly. A tracked-changes version and a clean revised manuscript have been submitted as requested.

Below, we provide a point-by-point response to all reviewer comments.

Reviewer #1

Comment 1

The introduction lacks a novelty statement. The last paragraph should focus on the aim and novelty of the work.

Response:

We thank the reviewer for this valuable comment. The Introduction has been revised to clearly articulate the novelty and scientific contribution of the study. A new paragraph has been added at the end of the Introduction highlighting (i) the use of neem-mediated green synthesis, (ii) integration of Ag into kaolin-derived zeolite X, and (iii) simultaneous evaluation of antimicrobial and antioxidant properties.

(Revised manuscript: Introduction, final paragraph)

Comment 2

Ensure consistent terminology for silver throughout the manuscript.

Response:

The manuscript has been thoroughly edited to ensure consistent terminology. “Silver,” “Ag⁺,” and “Ag-zeolite X” are now used consistently throughout the text depending on chemical context.

Comment 3

Correct scientific naming of microorganisms.

Response:

All microbial names have been corrected and standardized according to international nomenclature (italicized genus and species on first mention).

(Revised throughout the manuscript)

Comment 4

Remove repetition of DPPH and ABTS descriptions.

Response:

Redundant descriptions of DPPH and ABTS assays were removed, and the methodology was streamlined for clarity.

(Methods section: Antioxidant activity)

Comment 5

Provide accurate temperature and processing details for metakaolin preparation.

Response:

Detailed calcination conditions (temperature, duration, and heating procedure) have been added in the Materials and Methods section under “Synthesis of Zeolite X.”

Comment 6

Use correct botanical nomenclature for neem.

Response:

The scientific name Azadirachta indica has been used consistently at first mention and appropriately thereafter throughout the manuscript.

Comment 7

Expand the description of characterization techniques.

Response:

The descriptions of XRD, FTIR, SEM, and Raman spectroscopy have been significantly expanded, including instrumental conditions, scan ranges, and analytical rationale.

Comment 8

Clarify preparation of stock solutions in MIC assays.

Response:

The preparation of stock solutions, dilution steps, and concentrations have been clarified and standardized in (MIC determination).

Comment 9

Include JCPDS card numbers and literature support for XRD peaks.

Response:

Relevant JCPDS reference numbers and supporting literature have been added to the XRD results and discussion.

Comment 10

Add more peak assignments in FTIR and Raman analyses.

Response:

Additional FTIR and Raman peak assignments have been incorporated with corresponding literature citations, and interpretations were expanded to explain bonding and framework changes.

Comment 11

Improve Table 1 formatting and link antimicrobial results to SEM findings.

Response:

Table 1 has been reformatted for clarity. The antimicrobial results are now explicitly correlated with SEM-observed surface morphology and particle aggregation.

Comment 12

Correct p-value formatting.

Response:

All statistical values have been reformatted according to journal standards (e.g., p < 0.05).

Reviewer #2

Comment 1

Revise abstract to include key results.

Response:

The abstract has been rewritten to clearly summarize synthesis, characterization, antimicrobial and antioxidant outcomes, and key findings.

Comment 2

Strengthen novelty and justification in the Introduction.

Response:

The Introduction has been reorganized to emphasize the novelty of neem-assisted silver modification of zeolite X and its dual antimicrobial–antioxidant functionality.

Comment 3

Ensure uniformity in chemical and biological nomenclature.

Response:

All chemical names, microbial names, and abbreviations have been standardized throughout the manuscript.

Comment 4

Provide detailed characterization methodology.

Response:

Expanded experimental descriptions have been added for XRD, SEM, FTIR, Raman, and antimicrobial testing.

Comment 5

Include experimental temperature and time parameters.

Response:

All experimental conditions, including temperatures, durations, and concentrations, are now explicitly stated.

Comment 6

Strengthen discussion with literature support.

Response:

The Results and Discussion section has been substantially expanded with comparisons to relevant literature and mechanistic interpretations.

Comment 7

Add peak labeling in figures and scale bars in SEM images.

Response:

All figures have been updated with appropriate labels, peak assignments, and scale bars.

Comment 8

Include antimicrobial images in main manuscript.

Response:

Representative antimicrobial assay images have been included in the revised manuscript.

Comment 9

Clarify the novelty of the study.

Response:

The novelty is now explicitly stated in the Abstract, Introduction, and Discussion, emphasizing the green synthesis route, synergistic antimicrobial–antioxidant behaviour, and application relevance.

---

## [Decision Letter · Decision Letter 1]

2 Feb 2026

Antimicrobial and Antioxidant Activities of Neem assisted Silver-Modified Zeolite X Synthesized from Kaolin

PONE-D-25-58884R1

Dear Dr. Ralph Kwakye

We’re pleased to inform you that your manuscript has been judged scientifically suitable for publication and will be formally accepted for publication once it meets all outstanding technical requirements.

Kind regards,

Safdar Ali Amur

Academic Editor

PLOS One

Reviewers' comments:

Reviewer's Responses to Questions

**Comments to the Author**

Reviewer #1: All comments have been addressed

Reviewer #2: All comments have been addressed

2. Is the manuscript technically sound, and do the data support the conclusions?

Reviewer #1: Yes

Reviewer #2: Yes

3. Has the statistical analysis been performed appropriately and rigorously?

Reviewer #1: Yes

Reviewer #2: Yes

4. Have the authors made all data underlying the findings in their manuscript fully available?

Reviewer #1: Yes

Reviewer #2: Yes

5. Is the manuscript presented in an intelligible fashion and written in standard English?

Reviewer #1: Yes

Reviewer #2: Yes

Reviewer #1: All comments are answered by authors. All necessary changes have been done in MS. The current form of MS is acceptable. My recommendation is accepted for publication.

Reviewer #2: Authors have addressed all comments professionally and have made changes in MS. The current form of MS is acceptable, and it is recommended the study be published.

**Do you want your identity to be public for this peer review?** For information about this choice, including consent withdrawal, please see our Privacy Policy

Reviewer #1: No

Reviewer #2: No

---

## [Editor Report · Acceptance letter]

PONE-D-25-58884R1

PLOS One

Dear Dr. Kwakye,

I'm pleased to inform you that your manuscript has been deemed suitable for publication in PLOS One. Congratulations! Your manuscript is now being handed over to our production team.

Kind regards,

on behalf of

Safdar Ali Amur

Academic Editor

PLOS One